# Exploring COVID-19 Vaccination Willingness in Italy: A Focus on Resident Foreigners and Italians Using Data from PASSI and PASSI d’Argento Surveillance Systems

**DOI:** 10.3390/vaccines12020124

**Published:** 2024-01-26

**Authors:** Benedetta Contoli, Maria Elena Tosti, Federica Asta, Valentina Minardi, Giulia Marchetti, Virginia Casigliani, Salvatore Scarso, Silvia Declich, Maria Masocco

**Affiliations:** 1National Centre for Disease Prevention and Health Promotion, Istituto Superiore di Sanità, 00161 Rome, Italy; benedetta.contoli@iss.it (B.C.); valentina.minardi@iss.it (V.M.); maria.masocco@iss.it (M.M.); 2National Centre for Global Health, Istituto Superiore di Sanità, 00161 Rome, Italy; mariaelena.tosti@iss.it (M.E.T.); giulia.marchetti@iss.it (G.M.); salvatore.scarso@iss.it (S.S.); silvia.declich@iss.it (S.D.); 3Department of Translational Research and New Technologies in Medicine and Surgery, University of Pisa, 56126 Pisa, Italy; virginia.casigliani@phd.unipi.it; 4Department of Public Health and Infectious Diseases, Sapienza Università di Roma, 00185 Rome, Italy

**Keywords:** COVID-19 vaccination, vaccine hesitancy, surveillance systems, resident foreigners, public health strategies, sociodemographic determinants

## Abstract

(1) The COVID-19 pandemic exacerbated health disparities, both between foreign and autochthonous populations. Italy was one of the European countries that was the most affected by the COVID-19 pandemic; however, only limited data are available on vaccine willingness. This study aims to assess the propensity of foreign and autochthonous populations residing in Italy to be vaccinated and the relative associated factors. (2) Data were collected and analysed from the two Italian surveillance systems, PASSI and PASSI d’Argento, in the period of August 2020–December 2021. The data include those of the Italian resident adult population over 18 years old. A multinomial logistic regression model, stratified by citizenship, was used to assess the associations of sociodemographic, health, and COVID-19 experience variables with vaccination attitudes. (3) This study encompassed 19,681 eligible subjects. Considering the willingness to be vaccinated, foreign residents were significantly less certain to get vaccinated (49.4% vs. 60.7% among Italians). Sociodemographic characteristics, economic difficulties, and trust in local health units emerged as factors that were significantly associated with vaccine acceptance. Having received the seasonal flu vaccine was identified as a predictor of COVID-19 vaccine acceptance among foreign and Italian residents. (4) This study underscores the significance of tailoring interventions to address vaccine hesitancy based on the diverse characteristics of foreign and Italian residents. This research offers practical insights for public health strategies, highlighting the importance of tailored educational campaigns, improved communication, and nuanced interventions to enhance vaccine acceptance and uptake within both populations.

## 1. Introduction

After nearly three years, 766 million cases, and 6.9 million deaths [1], on 5 May 2023, the director of the World Health Organization (WHO), Tedros Adhanom Ghebreyesus, declared the end of COVID-19 as a public health emergency. However, as he pointed out, this does not mean that the disease is no longer a global threat. Moreover, the lessons learned during the emergency should serve to improve primary prevention and vaccination services with innovative approaches [2].

From a global health perspective, it is important to understand the impact of social inequalities on people’s health; in fact, many conditions of social vulnerability that increase disease risk and limit access to health services expose migrants to worse health outcomes [3]. In addition, migrant populations also face a higher risk of communicable diseases due to a limited access to preventive measures, such as vaccination [4,5]. Migrants face challenges in accessing vaccinations due to factors like low coverage in their home country, a lack of specific recommendations, administrative hurdles, limited resources, and socio-economic conditions. Tailored interventions like media campaigns and door-to-door vaccination initiatives have proven to be effective in overcoming these barriers [4,6,7].

The COVID-19 pandemic very clearly brought to light, and sometimes exacerbated, the health disparities that exist between migrant and indigenous populations [8,9,10,11,12,13].

Italy was one of the European Countries that was the most affected by the COVID-19 pandemic, reaching 6,270,000 cases (106,154 per 1,000,000 ab.) and 137,402 deaths (2329 per 1,000,000 ab.) in December 2021 according to the WHO COVID-19 Dashboard [14]. Several Italian studies showed worse outcomes in case of SARS-CoV-2 infection in migrant populations [15,16,17,18], even though, in some cases, a lower risk was highlighted, which was possibly due to a delayed diagnosis [16,17].

The observed disparities were often attributed to factors such as overcrowded facilities, a lower socioeconomic level, and language barriers that made all COVID-19 preventive interventions less effective among migrants. The need to implement targeted interventions tailored to reach migrants and ethnic minorities is evident [12].

In the context of COVID-19 vaccination, similar factors, including language barriers and difficulties in physical access, have been identified as the main reasons for reduced access to services [19]. In the same systematic review, which included data from 33 studies, mostly in high-income countries (HICs), the lack of confidence in the government and the system emerged as another important factor associated with poor communication. Trust in vaccine confidence was also found to have a role among black minorities in the UK [20].

In Italy, the few studies conducted at the local or regional levels that have investigated the coverage of COVID-19 vaccination among migrants showed a lower adherence in this group [17,21] and delayed access to vaccination compared to Italian citizens [22]. Additionally, comprehensive evidence on factors associated with vaccine acceptance and uptake within this population is lacking.

When addressing vaccination willingness, it is crucial to consider that vaccine hesitancy encompasses the delay in acceptance or outright refusal of vaccines, even when vaccination services are available. This hesitancy can vary over time, across different locations, among types of vaccines, and among various population subgroups [23]. A comprehensive understanding of vaccination hesitancy and specific attitudes among foreign residents is crucial for informed planning and the formulation of targeted interventions.

The objective of this study is to assess the propensity of the foreign resident population and autochthonous population in Italy to be vaccinated using data from Italian surveillance systems, namely PASSI (Progressi delle Aziende Sanitarie per la Salute in Italia–Progress in Local Health Units for Health in Italy) [24] and PASSI d’Argento [25], during the August 2020–December 2021 period.

This analysis can contribute valuable insights into public health strategies by investigating the factors influencing vaccine hesitancy in both autochthonous and foreign resident populations in Italy. Focus is placed on the foreign community.

## 2. Materials and Methods

### 2.1. PASSI and PASSI D’Argento Data Collections

PASSI is focused on the adult population (18–69-year-olds), collecting over 30,000 interviews annually; PASSI d’Argento is dedicated to people aged 65 years and over, collecting about 17,000 interviews annually. These surveillances cover approximately 90% of the Italian local health units (LHUs).

Both surveillance systems are based on the international framework of the US Centers for Disease Prevention and Control (CDC) [26,27] and mandated by the Italian Ministry of Health. In addition, they are officially acknowledged in accordance with the Decree of the Prime Minister’s Office on Registries and Surveillances, 3 March 2017 [28].

The National Institute of Health in Italy, referred to as Istituto Superiore di Sanità (ISS), is responsible for coordinating them at the national level. The operative hubs are the local health units (LHUs), where, on a monthly basis, specifically trained healthcare professionals conduct interviews with representative samples of the resident population [29,30].

PASSI and PASSI d’Argento collect information on several health-related topics, including health status, quality of life, behavioural risk factors, adherence to preventive programs, and specific aspects regarding “Active and Healthy Aging” policy for the elderly.

Between August 2020 and December 2021, a COVID-19 module was included into the standard questionnaire of both surveillance systems to investigate various COVID-19-related aspects, including perceptions, experiences with the infection, accessibility to care, mask use, emotional well-being, trust in the healthcare system’s ability to manage the emergency, economic impact, and the probable willingness to be vaccinated [31].

The exclusion criteria involved having a primary residence in another region, lacking a valid telephone number, being currently hospitalised, or residing in long-term care facilities, nursing homes, or prisons. People who do not speak Italian were also excluded, except in the autonomous province of Bolzano, where interviewees had the option of being interviewed in German.

Every selected individual received a notification letter from their local health unit (LHU) indicating that they would be contacted.

A monthly random sample was extracted from the resident lists in each LHU, stratified by sex and age group (18–34, 35–49, and 50–69 for PASSI; 65–74, 75−84, and ≥85 years of age for PASSI d’Argento) according to the population proportion in each stratum. The LHU data were merged, weighted (each record received a probability weight equivalent to the inverse of the sampling fraction within each LHU stratum), and analysed to obtain estimates at the national level. Additional methodological details can be found in previously published papers [24,25]. Throughout 2020 to 2021, the surveillance systems achieved a coverage rate of over 85% of the Italian population and a response rate of about 80% [32].

In Italy, registration is necessary to access the universally offered services of the National Health System (NHS) through the LHUs. In this study, the respondents residing in Italy with foreign citizenship were identified as foreign residents. Over 80% of the foreign sample has lived in Italy for more than ten years and has sufficient knowledge of Italian to answer the telephone survey, and this was one of the eligibility criteria for the interview.

Given that both surveillance systems share similar data collection methods and questionnaire items to investigate the analysed variables for the study, the two datasets from the PASSI and PASSI d’Argento surveillance systems were combined, considering interviewees aged 18–64 from PASSI and 65+ from PASSI d’Argento, obtaining a representative sample of the resident population in Italy aged 18 and older.

### 2.2. Outcome Definition and Study Variables

The primary aim of this study, assessing the propensity to get vaccinated, was evaluated through a question posed to the participants: “If a vaccine against COVID-19 were available, would you be willing to be vaccinated?” The respondents were provided with four possible answers: “definitely yes”, “probably yes”, “probably no”, and “definitely no”. Similarly to a previous research study conducted using data from the PASSI d’Argento surveillance system [33], the characteristics of the participants who responded “definitely yes”, “probably yes”, “probably no”, and “definitely no” were re-examined and redefined in terms of acceptance and hesitancy toward vaccination. Since no statistically significant differences between the “probably no” and “definitely no” groups were found, these two groups were combined into a single category labelled as refusal. On the contrary, the two groups of respondents who answered “definitely yes” and “probably yes” were treated individually and labelled as acceptance and inclination, respectively.

Therefore, in this study, acceptance to get vaccinated is indicated by the category of “definitely yes” respondents, while hesitancy to get vaccinated is indicated by the two categories named inclination (“probably yes”) and refusal (“probably no” and “definitely no”).

Demographic and socioeconomic factors, such us gender, age, education level (categorised as low for none or elementary/middle school for adults; low for none or elementary for the elderly; high for high school or university for adults; and high for middle school, high school, or university for the elderly), economic difficulties in making ends meet with the available financial resources (yes, no), and geographic residence area categorised by the National Institute of Statistics criteria (North, Centre, South, and major islands), were examined in relation to the probable willingness to receive the COVID-19 vaccination.

Other conditions were examined, including those related to the following health statuses: influenza vaccination during the previous year (yes vs. no) registered for all respondents (in Italy, flu vaccination is not mandatory, but only recommended for people at higher risk, such as elderly people (65+), adults with chronic pathologies, people who live with or take care of people at higher risk, pregnant women, obese people, healthcare workers, and the military) and the presence of self-reported chronic diseases (at least one vs. none), including one or more of the following pathologies: diabetes, cancer, chronic bronchitis, kidney failure, asthma, emphysema, stroke myocardial infarction, respiratory insufficiency, coronary and other heart diseases, and chronic liver disease or cirrhosis.

Lastly, considerations were given to the year of the COVID-19 pandemic (2021 vs. 2020), as well as attitudes and experiences related to COVID-19. These included the perceived risk of infection for oneself or one’s family (high vs. low); trust in the local health unit (LHU) to manage the COVID-19 situation (yes vs. no); occurrences of COVID-19 cases among family, friends, or colleagues (yes vs. no); the unfortunate event of the death of relatives or friends due to COVID-19 infection (yes vs. no); concerns about the current situation (yes vs. no); and the presence of intrusive thoughts, signifying recurring and distressing thoughts associated with experiences during the health emergency (yes vs. no).

Since the aim of this paper is to understand the willingness to receive the COVID-19 vaccine and associated factors, only people who completed the COVID-19 module were eligible for the analysis.

The flowchart in Figure 1 shows details regarding the individuals who were excluded from the analysis and the reasons behind their exclusion. Interviews with a proxy (a person close to the older that supported him/her during the interview) or those that were interrupted; interviews with “I had COVID-19”, “I had already vaccinated”, “I don’t know” or missing values as answers to the question “If a vaccine against COVID-19 were available, would you be willing to be vaccinated?”; and interviews with missing information on citizenship were excluded from the analysis. Nevertheless, a specific analysis of the characteristics of the interviewees who reported “I don’t know” was performed, and it showed that these answers came from a slightly higher proportion of women, from people reporting a lower level of education, or from people with economic difficulties in making ends meet.

### 2.3. Statistical Analysis

Results from descriptive analysis are presented as percentages with corresponding 95% confidence intervals (CIs). To examine the associations of sociodemographic factors, health indicators, and COVID-19 experiences with vaccination attitudes among both foreign and Italian residents, univariate and multivariate multinomial logistic regression analyses were conducted separately. In the multinomial regression model, the acceptance group represented the reference population; the outcome variable was coded to compare the acceptance group with both the inclination and refusal groups. The results are presented as relative risk ratios (RRRs) with corresponding 95% confidence intervals (CIs). The RRRs compared the risk of the outcome (e.g., inclination and refusal) among one group compared with a reference group (e.g., females compared with males, where males serve as the reference group) [34]. All data were analysed using STATA, version 16.0 [35].

## 3. Results

### 3.1. Sample Characteristics

During the study period, 19,681 eligible subjects were interviewed within the two surveillance systems; 18,741 were Italian citizens (95.2%), and 940 were foreign-born (4.8%) (Figure 1). All analysed variables, including sociodemographic factors, health status, and factors related to COVID-19 attitudes and experiences of the study population, are presented in Table 1.

In the overall population and when examining Italian and foreign residents separately, the number of females exceeded that of males (51.4%, 51.1%, and 57.2%, respectively). The Italian residents were, on average, 49 years old, while the foreigners were slightly younger (43 years old). The majority of the Italian sample (71.6%) had a high educational level; in the foreign sample, this percentage significantly decreased to 58.1%. Almost half of the Italian residents, accounting for 50.6%, resided in the south of Italy, whereas only 12.1% of foreigners lived in that region.

The majority of both samples declared that they did not receive the seasonal flu vaccine in the past 12 months (75.8% vs. 90.0%, respectively). About 24% of Italian residents had at least one non-communicable chronic disease, while in the foreign sample, this value was only 17.1%.

Overall, less than half (46.8%) of the participants thought that the probability of contracting COVID-19 was high (47.2% for Italians and 37.6% for foreigners); in the Italian sample, about 30% had experienced intrusive thoughts, while in the foreign sample, this percentage decreased to 27.0%; and most of the respondents from both populations had trust in their LHUs to manage the COVID-19 situation (75.4% in Italians and 77.6% in foreigners).

Furthermore, considering the willingness to be vaccinated, the foreign residents were significantly less certain to be vaccinated (49.4% vs. 60.7% among Italians) and more frequently reported “definitely no” (12.0% vs. 6.9% among Italians) or “probably no” (15.1% vs. 9.1% among Italians). The percentages were similar for “probably yes” (Table 1).

### 3.2. Regression Model Results

We tested the relation of citizenship with the willingness to get vaccinated for COVID-19 in a multinomial model adjusted for demographic and socioeconomic variables as well as those related to health status, and we found a significant association with the refusal group (RRR = 1.34; 95% CI 1.07–1.68).

The results of the multinomial logistic regression analysis for hesitancy to get vaccinated, which stands for the inclination and refusal groups, versus acceptance to get vaccinated among the Italian sample are presented in Table 2.

The Italian inclination group was characterised by the presence of economic difficulties (RRR = 1.57; 95% CI 1.37–1.80) and by being worried about the emergency (RRR = 1.24; 95% CI 1.06–1.44); in the refusal group, the presence of economic difficulties (RRR = 1.64; 95% CI 1.38–1.95) and distrust in their LHU’s capacity to manage the COVID-19 situation (RRR = 2.15; 95% CI 1.82–2.54) presented the strongest associations.

People with high education levels had lower likelihoods to hesitate receiving the vaccine (RRR = 0.76 and 95% CI 0.66–0.88 for inclination and RRR = 0.52 and 95% CI 0.43–0.61 for refusal).

Moreover, people who declared COVID-19 cases among relatives or friends had lower likelihoods to refuse the vaccine (RRR = 0.69; 95% CI 0.59–0.80).

The results of the multinomial logistic regression analysis for hesitancy (both inclination and refusal) versus acceptance among foreign residents are shown in Table 3.

The foreign inclination group had concerns about the current situation regarding COVID-19 (RRR = 1.87; 95% CI 1.07–3.28). The most important risk factor associated with the refusal to receive a COVID-19 vaccination in the foreign group was being female (RRR = 1.68; 95% CI 1.01–2.79). Other risk factors such as a low educational level, economic difficulties, trust in LHUs’ management capacity, or the presence of COVID-19 cases among family, friends, or colleagues presented an RRR higher than one but did not reach the statistical significance due to the size of the sample.

Another important result to note, which is valid for all groups (Table 2 and Table 3), is that people who received the seasonal flu vaccine had a lower willingness to hesitate receiving vaccination against COVID-19 (Italians: RRR = 0.44 and 95% CI 0.37–0.51 and RRR = 0.13 and 95% CI 0.10–0.18, respectively, for inclination and refusal; foreigners: RRR = 0.44, 95% CI 0.21–0.93 and RRR= 0.10, 95% CI 0.03–0.34, respectively, for inclination and refusal). Furthermore, in 2021, compared to 2020, the hesitancy to get vaccinated against COVID-19 decreased in all four subgroups (Italians: RRR = 0.24 and 95% CI 0.21–0.28 and RRR = 0.33 and 95% CI 0.29–0.39, respectively, for inclination and refusal; foreigners: RRR = 0.42 and 95% CI 0.25–0.70 and RRR = 0.29 and 95% CI 0.18–0.48, respectively, for inclination and refusal).

## 4. Discussion

This study aimed to assess the propensity of foreign residents in Italy to get vaccinated against COVID-19 compared to the autochthonous population during the 2020–2021 period. This research highlighted differences in vaccination willingness and explored the possible determinants of these differences between foreign and Italian residents. Notably, the data collected in 2020 predate the initiation of the vaccination campaign in Italy on 27 December 2020, known as “Vaccine Day”. In the subsequent phase in 2021, though vaccines were available, initial access was limited to people in fragile categories [36,37].

The study sample was extracted from the resident population in Italy during the study period, revealing sociodemographic disparities between Italian and foreign residents. The foreigners tended to be younger, possess lower educational levels, predominantly reside in North Italy, and face higher economic difficulties compared to Italians. These findings align with the expected characteristics of the foreign population in Italy [38]. Moreover, the foreign sample exhibited lower non-communicable chronic diseases.

Despite migrants being overrepresented among COVID-19 cases [12,16] and experiencing an increased infection risk and adverse outcomes during the pandemic [13], our study did not demonstrate a significant difference in the reported impact on family and friends between Italians and foreigners. In fact, our study highlighted that a similar percentage of Italians and foreigners report to have experienced deaths among families and friends and a smaller proportion of foreigners reported COVID-19 cases among relatives, friends, and colleagues; moreover, regarding the attitudes and experiences related to COVID-19, foreign residents reported less concern and fewer intrusive thoughts about the pandemic situation. However, underdiagnoses and delayed diagnoses of SARS-CoV-2 in non-Italian nationals [15] might contribute to their different perceptions compared to Italians.

Previous studies conducted in Italy on the influenza vaccination showed a lower vaccination coverage among migrants (17% for influenza in elderly and adults at risk) compared to Italian citizens (40%) [39]. Similarly, in this study, while the majority of foreign residents expressed a definite willingness to get vaccinated (“definitely yes”—49.4%) versus the other categories, “probably yes” (23.5%), “probably no” (15.1%), and “definitely no” (12.0%), this percentage was still significantly lower than that of Italians (49.4% vs. 60.7%).

A multivariate analysis revealed that sociodemographic factors influenced vaccination propensity; notably, there was a (female) gender-related refusal of the COVID-19 vaccination among the foreign resident group, unlike the Italian group. No significant association was reported in scientific literature between gender and migrants’ willingness to get vaccinated [40,41,42], while two systematic reviews reported that the female gender is a determinant for hesitancy in the Italian population [43,44].

Economic difficulties were significantly associated with vaccine hesitancy (both inclination and refusal) in the Italian group, while this association was observed at the threshold of significance among the foreign residents. Previous Italian studies about the willingness of people residing in Italy to receive the COVID-19 vaccination (both autochthonous and foreign residents) have highlighted that hesitant people were more likely to have economic difficulties, lower incomes, or be unemployed [45,46]. Another study related to the COVID-19 vaccine hesitancy of parents of adolescents between 12 and 17 years of age in Italy reported that the hesitant/reluctant parents had low incomes [47].

The study also highlighted that health-status-related factors are significantly associated with an increased willingness to get vaccinated in both the foreign and Italian populations, such as those who received the seasonal flu vaccine. This is in line with a previous study using data from the PASSI d’Argento surveillance system in the elderly resident population [33].

A multivariate analysis revealed an overall increase in the willingness to get vaccinated against COVID-19 in 2021 versus 2020 for both Italian and foreign residents. The factors influencing this trend included the initiation of the vaccination campaign, increased knowledge about vaccines, changes in disease dynamics, and other contextual changes such as expanded vaccination categories and the introduction of the green pass [37,48,49,50].

Both Italian and foreign residents demonstrated a significant correlation between vaccine inclination and concern about the current COVID-19 situation, with heightened worry decreasing the likelihood of vaccine refusal. An Australian study examining the willingness to receive a COVID-19 booster vaccine among migrants from the Eastern Mediterranean found that they shared similar perceptions of personal risks from COVID-19. However, those willing to receive a booster viewed COVID-19 as a broader health issue [42]. Additionally, a review indicated that the perception of risks associated with COVID-19 (such as the severity of the illness, high exposure, and susceptibility to infection) is linked to the acceptance of vaccination [44].

As expected, distrust in the management capacity of LHUs was a significant factor for the refusal of vaccination in the Italian group and was associated with a limited significance among foreign residents, which was possibly due to poor communication regarding vaccines, disease outcomes, challenges in contact tracing, new variants, and varying vaccine efficacy. A larger sample size might yield significant results. Similar findings concerning distrust in institutions, the government, public health authorities, and in the information received were reported in various studies on the willingness to get vaccinated [40,41,42,43,44,51,52].

Apart from the factors investigated in the present study that impact the inclination to get vaccinated against COVID-19, other studies focusing on migrant populations have highlighted additional considerations. These include apprehensions concerning the vaccination safety and potential side effects (not only COVID-19-related) or worrying about the news about COVID-19 vaccines [41,42,51]. Similar concerns have been observed in studies conducted within Italy as well [44]. In addition, it could be important to take into account health literacy as a determinant of influenza vaccination status. Even if in a regional study conducted in Italy, health literacy did not seem to affect the likelihood of influenza vaccination uptake, it might be relevant to explore this association in depth with ad hoc studies.

### Limitations and Strengths

This study presents some limitations. Firstly, the self-reported data may be susceptible to distortions (due to recall bias or social desirability), potentially leading to an underestimation or overestimation of the phenomena under investigation. However, numerous studies have demonstrated that self-reported data from the American Behavioural Risk Factor Surveillance System, which PASSI and PASSI d’Argento draw inspiration from (for study design and data collection and to analyse the methodology), exhibit robust reliability and validity [53]. Secondly, there is an underestimation of the foreigner sample from PASSI and PASSI d’Argento, amounting to 5%, whereas according to the National Statistical Institute (ISTAT), this number would be 8% of residents in Italy on 1 January 2021 [54]. This underestimation could be explained by the eligibility criteria that required sufficient knowledge of the Italian language to answer the telephone survey. This inclusion criteria could make the foreigner sample considered in the study not fully representative of all foreigners residing in Italy, but of the well-integrated ones. Moreover, due to the limited sample size of foreigners, no data on the country of origin was used in this analysis, even though this factor had an association with COVID-19 impact and vaccine coverage in recent Italian studies [16,55].

Despite these limitations, this study has significant strengths. Primarily, the results were derived from a large gender- and age-representative sample of the population residing in Italy from 18 years onward. Secondly, it includes data derived from two surveillance systems which share the same methodology (selection criteria, sampling, collecting, and analysing data instruments and operative protocol).

Finally, a big strength of this study, which is important to highlight, is the opportunity provided by surveillance systems to use socio-demographic characteristics and health-related conditions as confounders in the study of the willingness to get vaccinated based on citizenship.

## 5. Conclusions

Our study showed significant differences between Italian and foreign residents in the willingness to receive the COVID-19 vaccine and in the associated factors.

Considering the eligibility criteria of the study population, it is possible to assume that foreign residents included in the study are likely well integrated or have already started the integration process; moreover, it is plausible that they are familiar with the National Healthcare System and engage with health promotion programs. Despite this, significant differences in vaccination propensity between foreign and Italian residents emerged from the study. We therefore expect these differences in attitude to be even more pronounced between Italians and non-Italian speakers or newly arrived foreigners present in Italy.

The analysis of the determinants of vaccine hesitancy in both groups (Italian and foreign residents) enables us to highlight possible fields of intervention, and our results suggest the necessity for tailored interventions and educational programs to promote vaccine literacy, dispel concerns and misinformation, and boost vaccine confidence and uptake among migrants [2]. In particular, since socio-economic determinants are significantly associated with vaccine hesitancy in Italian residents and at the limit of significance in foreigners, this highlights the need to improve vaccine offer by making vaccination services and more health services in general that are more inclusive and reachable to even the most disadvantaged segments of the population. If the health system were able to be more accessible to all individuals, this could contribute to contrast the distrust toward health institutions.

From a public health perspective, understanding the characteristics and perceptions of vaccine-hesitant individuals is crucial to plan and implement effective immunisation promotion strategies. The findings of this study could inform public policies aimed at enhancing immunisation access and reducing inequalities in public health.

## Figures and Tables

**Figure 1 vaccines-12-00124-f001:**
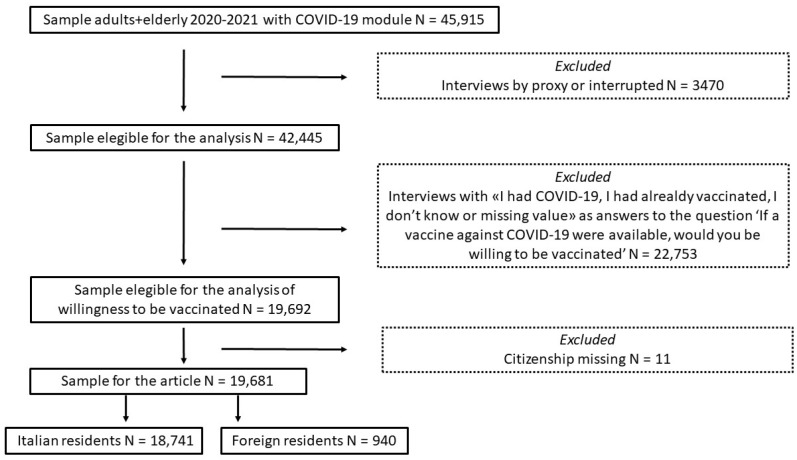
Flow chart describing the selection procedure for subsamples from adults plus elderly respondents with COVID-19 module of the PASSI and PASSI d’Argento surveillance systems, 2020–2021.

**Table 1 vaccines-12-00124-t001:** Characteristics of adult and elderly residents in Italy overall and based on citizenship according to PASSI and PASSI d’Argento, 2020–2021.

		All		Italian Citizenship (*n* = 18,741)		Foreign Citizenship (*n* = 940)	
		*n*	%	%	IC 95%	%	IC 95%
**Total**		19,681	100.0	94.8	94.4–95.2	5.2	4.8–5.6
**Sociodemographic factors**							
Gender	Male	9317	48.6	48.9	48.4–49.4	42.8	40.3–45.4
	Female	10,363	51.4	51.1	50.6–51.6	57.2	54.6–59.6
Age	Mean (sd)	48.8 (0.10)	-	49.2 (0.10)	-	43.0 (0.37)	-
Educational level	Low	5848	29.0	28.3	27.4–29.3	41.9	38.0–46.0
	High	13,785	71.0	71.6	70.7–72.5	58.1	54.0–62.0
Economic difficulties	Yes	7459	40.2	39.2	38.2–40.2	58.2	54.3–60.0
	No	12,096	59.8	60.8	59.8–61.8	41.8	38.0–45.7
Geographic area of residence	North	7232	32.8	31.2	30.8–31.6	61.8	59.4–64.1
	Centre	3991	18.6	18.2	17.9–18.4	26.1	24.5–27.8
	South	8457	48.6	50.6	50.1–51.1	12.1	10.1–14.5
Seasonal flu vaccination uptake	Yes	5401	23.4	24.2	23.4–25.0	10.0	8.2–12.3
	No	14,251	76.6	75.8	75.0–76.5	90.0	87.7–91.9
Period	2020	10,023	53.5	53.6	52.8–54.3	51.7	47.8–55.5
	2021	9657	46.5	46.4	45.7–47.1	48.3	44.5–52.2
Noncommunicable chronic diseases	None	14,397	76.1	75.8	74.8–76.7	82.9	79.9–85.6
	At least one	5273	23.9	24.2	23.3–25.2	17.1	14.4–20.1
**COVID-19 attitudes and experiences**							
Probability of infection of SARS-Cov-2	High	7364	46.76	47.2	46.2–48.3	37.6	33.3–42.1
	Low	9686	53.24	46.3	45.3–47.4	48.7	44.2–53.2
Reported intrusive thoughts	Yes	5313	29.6	29.7	28.8–30.6	27.0	23.6–30.7
	No	13,755	70.4	70.3	69.4–71.2	73.0	69.3–76.4
Reported being worried ^1^	Yes	13,936	71.9	72.5	71.5–73.5	60.3	56.2–64.2
	No	5491	28.1	27.5	26.5–28.5	39.7	35.8–43.8
COVID-19 deaths in family or friends	Yes	1262	6.3	6.3	5.8–6.8	6.2	4.5–8.5
	No	18,375	93.7	93.7	93.2–94.2	93.8	91.5–95.5
Trust in local health unit management	Yes	13,040	75.5	75.4	74.3–76.3	77.6	73.7–81.0
	No	4047	24.5	24.7	23.6–25.7	22.4	19.0–26.3
COVID-19 cases in family, friends, or colleagues	Yes	10,230	49.7	50.1	49.1–51.2	42.2	38.2–46.3
	No	9239	50.3	49.9	48.8–50.9	57.8	53.7–61.8
Willingness to get vaccinated	Definitely yes	12,063	60.1	60.7	60.0–61.7	49.4	45.5–53.4
	Probably yes	4417	23.3	23.3	22.5–24.2	23.5	20.3–27.1
	Probably no	1795	9.4	9.1	8.5–9.7	15.1	12.4–18.2
	Definitely no	1406	7.2	6.9	6.4–7.5	12.0	9.5–15.1

^1^ About the emergency.

**Table 2 vaccines-12-00124-t002:** Crude and adjusted Relative Risk Ratio of multinomial analysis for sociodemographic and COVID-19-related risk factors associated with hesitancy to accept the COVID-19 vaccine among Italian residents according to PASSI and PASSI d’Argento, 2020–2021.

		Italian Residents *n* = 18,741							
		Multinomial Univariate				Multinomial Multivariate			
		Hesitancy				Hesitancy			
		Inclination versus Acceptance		Refusal versus Acceptance		Inclination versus Acceptance		Refusal versus Acceptance	
Characteristics		RRR *	95% CI	RRR *	95% CI	RRR *	95% CI	RRR *	95% CI
**Sociodemographic factors**									
Gender	Female (ref. male)	0.96	(0.87–1.06)	1.05	(0.94–1.18)	0.96	(0.85–1.08)	1.10	(0.94–1.27)
Age		**0.99**	**(0.98–0.99)**	**0.99**	**(0.98–0.99)**	1.00	(0.99–1.00)	1.00	(0.99–1.01)
Educational level	High (ref. low)	**0.81**	**(0.73–0.91)**	**0.53**	**(0.47–0.60)**	**0.76**	**(0.66–0.88)**	**0.52**	**(0.43–0.61)**
Economic difficulties	Yes (ref. no)	**1.52**	**(1.37–1.69)**	**1.63**	**(1.45–1.83)**	**1.57**	**(1.37–1.80)**	**1.64**	**(1.38–1.95)**
Geographic area of residence	Centre (ref. North)	**0.82**	**(0.72–0.92)**	**0.57**	**(0.49–0.66)**	1.09	(0.93–1.27)	**0.68**	**(0.56–0.83)**
	South (ref. North)	**0.89**	**(0.80–0.98)**	**0.60**	**(0.53–0.68)**	**0.77**	**(0.67–0.89)**	**0.50**	**(0.42–0.60)**
Seasonal flu vaccination uptake	Yes (ref. no)	**0.48**	**(0.43–0.54)**	**0.17**	**(0.14–0.20)**	**0.44**	**(0.37–0.51)**	**0.13**	**(0.10–0.18)**
Period	2021 (ref. 2020)	**0.28**	**(0.25–0.31)**	**0.46**	**(0.41–0.51)**	**0.24**	**(0.21–0.28)**	**0.33**	**(0.29–0.39)**
Noncommunicable chronic diseases	At least one (ref. none)	0.75	(0.66–0.84)	0.87	(0.75–1.00)	0.91	(0.77–1.06)	1.14	(0.92–1.40)
**COVID-19 attitudes and experiences**									
Probability of infection of SARS-CoV-2	High (ref. low)	**1.14**	**(1.02–1.27)**	**0.72**	**(0.63–0.82)**	0.99	(0.87–1.12)	**0.83**	**(0.70–0.97)**
Reported intrusive thoughts	Yes (ref. no)	**0.76**	**(0.67–0.85)**	**0.74**	**(0.64–0.85)**	**0.80**	**(0.69–0.92)**	0.88	(0.72–1.07)
Reported being worried ^1^	Yes (ref. no)	**1.42**	**(1.25–1.61)**	**0.65**	**(0.57–0.74)**	**1.24**	**(1.06–1.44)**	**0.62**	**(0.52–0.73)**
Trust in management capacity of local health unit	No (ref. yes)	1.13	(0.99–1.30)	**2.12**	**(1.84–2.44)**	1.08	(0.93–1.26)	**2.15**	**(1.82–2.54)**
COVID-19 deaths in family or friends	Yes (ref. no)	**0.64**	**(0.52–0.79)**	**0.61**	**(0.47–0.78)**	0.78	(0.60–1.01)	0.83	(0.60–1.15)
COVID-19 cases among family, friends, or colleagues	Yes (ref. no)	**0.78**	**(0.71–0.87)**	**0.65**	**(0.58–0.73)**	0.92	(0.81–1.04)	**0.69**	**(0.59–0.80)**

^1^ About the emergency. * Significant findings are in bold.

**Table 3 vaccines-12-00124-t003:** Crude and adjusted Relative Risk Ratio for sociodemographic and COVID-19-related risk factors associated with hesitancy to accept the COVID-19 vaccine among foreign residents according to PASSI and PASSI d’Argento, 2020–2021.

		Foreign Residents *n* = 940							
		Multinomial Univariate				Multinomial Multivariate			
		Hesitancy				Hesitancy			
		Inclination versus Acceptance		Refusal versus Acceptance		Inclination versus Acceptance		Refusal versus Acceptance	
Characteristics		RRR *	95% CI	RRR *	95% CI	RRR *	95% CI	RRR *	95% CI
**Sociodemographic factors**									
Gender	Female (ref. male)	1.16	(0.77–1.74)	1.48	(0.99–2.21)	1.07	(0.62–1.83)	**1.68**	**(1.01–2.79)**
Age		1.00	(0.98–1.01)	1.00	(0.99–1.02)	1.00	(0.98–1.02)	1.01	(0.99–1.03)
Educational level	High (ref. low)	0.83	(0.55–1.25)	**1.57**	**(1.04–2.36)**	0.75	(0.44–1.28)	1.30	(0.76–2.21)
Economic difficulties	Yes (ref. no)	1.54	(0.99–2.37)	1.16	(0.78–1.73)	1.69	(0.97–2.95)	1.63	(0.96–2.75)
Geographic area of residence	Centre (ref. North)	0.87	(0.58–1.30)	**0.50**	**(0.32–0.76)**	1.07	(0.63–1.81)	0.64	(0.3511–1.18)
	South (ref. North)	0.60	(0.33–1.09)	0.67	(0.36–1.25)	0.75	(0.40–1.42)	0.53	(0.25–1.13)
Seasonal flu vaccination uptake	Yes (ref. no)	0.66	(0.38–1.15)	**0.25**	**(0.12–0.52)**	**0.44**	**(0.21–0.93)**	**0.10**	**(0.03–0.34)**
Period	2021 (ref. 2020)	**0.38**	**(0.25–0.57)**	**0.35**	**(0.24–0.52)**	**0.42**	**(0.25–0.70)**	**0.29**	**(0.18–0.48)**
Noncommunicable chronic diseases	At least one (ref. none)	0.89	(0.52–1.50)	0.90	(0.56–1.46)	0.88	(0.44–1.75)	1.21	(0.63–2.29)
**COVID-19 attitudes and experiences**									
Probability of infection of SARS-CoV-2	High (ref. low)	1.23	(0.79–1.90)	**0.54**	**(0.33–0.88)**	1.07	(0.63–1.82)	**0.56**	**(0.32–0.97)**
Reported intrusive thoughts	Yes (ref. no)	0.84	(0.53–1.30)	**0.52**	**(0.33–0.83)**	0.67	(0.37–1.21)	0.63	(0.33–1.19)
Reported being worried ^1^	Yes (ref. no)	**2.07**	**(1.34–3.17)**	**0.65**	**(0.43–0.97)**	**1.87**	**(1.07–3.28)**	**0.51**	**(0.30–0.88)**
Trust in management capacity of local health unit	No (ref. yes)	**1.78**	**(1.03–3.06)**	**1.98**	**(1.16–3.38)**	1.59	(0.85–2.95)	1.79	(0.96–3.35)
COVID-19 deaths in family or friends	Yes (ref. no)	0.94	(0.42–2.10)	**0.33**	**(0.12–0.92)**	1.04	(0.37–2.93)	0.46	(0.12–1.80)
COVID-19 cases among family, friends, or colleagues	Yes (ref. no)	1.36	(0.90–2.04)	0.931	(0.62–1.39)	1.35	(0.77–2.33)	1.33	(0.78–2.29)

^1^ About the emergency. * Significant findings are in bold.

## Data Availability

The data presented in this study are available upon request from the corresponding author. The data are not publicly available due to privacy restrictions.

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
