# Peer review of "Exploring COVID-19 Vaccination Willingness in Italy: A Focus on Resident Foreigners and Italians Using Data from PASSI and PASSI d’Argento Surveillance Systems"

_vaccines, 2024, doi:10.3390/vaccines12020124_

Round 1
Reviewer 1 Report
Comments and Suggestions for Authors
The study titled 'Exploring COVID-19 Vaccination Willingness in Italy: A Focus on Resident Foreigners and Italians Using Data from Surveillance Systems PASSI and PASSI d’Argento' is an in-depth study assessing the propensity to be vaccinated against COVID-19 among foreign and Italian residents in Italy. It utilizes data from the Italian surveillance systems PASSI and PASSI d’Argento, covering the period from August 2020 to December 2021.
Here are some suggestions for improvement and points of consideration:
- Introduction: The introduction is quite wordy and somewhat out of focus compared to the title. I recommend providing a very brief background on Italy's COVID-19 situation and then focusing specifically on what is known about the vaccination willingness among different demographic groups in Italy, highlighting the importance of comparing resident foreigners and Italians in understanding public health responses.
- The methodology section of the manuscript could be enhanced by providing more detailed information. E.g., exact dates of data collection are missing (in case exact dates are august 2020 - december 2021, please comment on the fact that so many things changed during this tame range during the pandemic).
(line 123): What do you mean by 'acceptable precision?'
While the paper explains that data were collected from the PASSI and PASSI d'Argento surveillance systems, readers would benefit from a more detailed explanation of the sampling procedure.
The exclusion of non-Italian speakers (except in Bolzano) might introduce a significant bias in the results. It's crucial to discuss how this limitation impacts the findings, especially given the focus on resident foreigners. It limits the generalizability of the findings, which may only represent foreigners that are long-term, well-integrated in Italy.
- Results: Including previous flu vaccination in the statistical models can be somewhat misleading, assuming you also included part of the population for which this vaccine was not recommended.
- Discussion: In the discussion section, while you link the findings to broader public health implications, there could be more in-depth exploration of how these insights specifically inform public health strategies in Italy. This would make the discussion more practical and relevant.
In this manuscript, you haven't analyzed the potential role of health or vaccine literacy. Do you deem these as relevant factors that might influence or moderate willingness to get vaccinated or take appropriate decisions when it comes to vaccinations? I suggest commenting on it in the discussion section.
Comments on the Quality of English Language
Minor editing required. English is good overall.
Author Response
Review 1
(x) I would not like to sign my review report
( ) I would like to sign my review report
Quality of English Language
( ) I am not qualified to assess the quality of English in this paper
( ) English very difficult to understand/incomprehensible
( ) Extensive editing of English language required
( ) Moderate editing of English language required
(x) Minor editing of English language required
( ) English language fine. No issues detected
|
Yes |
Can be improved |
Must be improved |
Not applicable |
|
|
Does the introduction provide sufficient background and include all relevant references? |
( ) |
( ) |
(x) |
( ) |
|
Are all the cited references relevant to the research? |
( ) |
(x) |
( ) |
( ) |
|
Is the research design appropriate? |
( ) |
(x) |
( ) |
( ) |
|
Are the methods adequately described? |
( ) |
( ) |
(x) |
( ) |
|
Are the results clearly presented? |
( ) |
( ) |
(x) |
( ) |
|
Are the conclusions supported by the results? |
( ) |
(x) |
( ) |
( ) |
Comments and Suggestions for Authors
The study titled 'Exploring COVID-19 Vaccination Willingness in Italy: A Focus on Resident Foreigners and Italians Using Data from Surveillance Systems PASSI and PASSI d’Argento' is an in-depth study assessing the propensity to be vaccinated against COVID-19 among foreign and Italian residents in Italy. It utilizes data from the Italian surveillance systems PASSI and PASSI d’Argento, covering the period from August 2020 to December 2021.
Here are some suggestions for improvement and points of consideration:
- Introduction: The introduction is quite wordy and somewhat out of focus compared to the title. I recommend providing a very brief background on Italy's COVID-19 situation and then focusing specifically on what is known about the vaccination willingness among different demographic groups in Italy, highlighting the importance of comparing resident foreigners and Italians in understanding public health responses.
R: According to the referee comment, the introduction was shortened. We also added numbers of cases and death in Italy from WHO source that provide the dimension of the emergency in Italy at December 2021 (referring to our study period 2020-2021).
In lines 73-75, we already reported a sentence on what is known about vaccination willingness, but, as underlined: “ comprehensive evidence on factors associated with vaccine acceptance and uptake within this population is lacking”.
Moreover, we tried to highlight the possible contribution of this study in a public health prospective. In particular, we better specified that we intended to investigate both populations (autochthonous and foreign residents in Italy) with a focus on the foreign community.
- The methodology section of the manuscript could be enhanced by providing more detailed information. E.g., exact dates of data collection are missing (in case exact dates are august 2020 - december 2021, please comment on the fact that so many things changed during this tame range during the pandemic).
R: According to the reviewer suggestion, we added a phrase on the exact dates of data collection in the Materials and Methods section, paragraph 2.1. PASSI and PASSI D'Argento data collection (Pag: 3, line: 108). Moreover, in the discussion section we already commented factors influencing changes during the pandemic period including the initiation of the vaccination campaign, increased knowledge about vaccines, changes in disease dynamics, and other contextual changes such as expanded vaccination categories and the introduction of the green pass.
(line 123): What do you mean by 'acceptable precision?'
R: Thank you for your comment, we decide to delete the related phrase on the paper, since a single sentence is not sufficient, as the concept requires a specific paragraph that can be found in references 25 and 26. However, we added a phrase to better clarify (Pag: 3, line: 122-125).
In brief, for each LHUs level the sample size was sufficient to estimate prevalences of several health-related topics with an a priori defined precision (+/- 5% of the prevalence) giving a robust evaluation of the phenomenon.
While the paper explains that data were collected from the PASSI and PASSI d'Argento surveillance systems, readers would benefit from a more detailed explanation of the sampling procedure.
R: Thanks for your suggestion, as written in the paper, the sample is randomly selected from the LHUs list of beneficiaries of health services of the participating regions, stratified by sex and age. Since there were many articles based on these data and we cited references 25 and 26 based on methodological issues, we think it could be redundant to add more detailed explanation of the sampling procedure.
The exclusion of non-Italian speakers (except in Bolzano) might introduce a significant bias in the results. It's crucial to discuss how this limitation impacts the findings, especially given the focus on resident foreigners. It limits the generalizability of the findings, which may only represent foreigners that are long-term, well-integrated in Italy.
R: Eligibility criteria required sufficient knowledge of Italian to answer the telephone survey. So it is possible to assume that foreign residents included in the study are well integrated or have already started an integration process. This could make the foreigners selected for the study not fully representative of all foreigners residing in Italy. To better clarify the generalizability of the findings, we added a phrase in the 4.1 paragraph (Pag: 11, line: 375-377) and another comment in the conclusion section (Pag: 12, line: 397-398).
- Results: Including previous flu vaccination in the statistical models can be somewhat misleading, assuming you also included part of the population for which this vaccine was not recommended.
R: We had information on flu vaccination for all sample (18+), for this reason we did not exclude anyone in the statistical models. In Italy flu vaccination is not mandatory but only recommended for people at higher risk: elderly people (65+), adults with chronic pathologies, people who live with or care people at higher risk, pregnant women, obese, healthcare workers and military.
To better clarify we added the previous sentences in the paper (Pag: 4, line: 164-167).
- Discussion: In the discussion section, while you link the findings to broader public health implications, there could be more in-depth exploration of how these insights specifically inform public health strategies in Italy. This would make the discussion more practical and relevant.
R: According to the referee comment, the public health implications were improved with few sentences in the conclusion section (Pag: 12, line: 399-409).
In this manuscript, you haven't analyzed the potential role of health or vaccine literacy. Do you deem these as relevant factors that might influence or moderate willingness to get vaccinated or take appropriate decisions when it comes to vaccinations? I suggest commenting on it in the discussion section.
R: Thank you for the suggestion, we added in the discussion section of the paper a sentence on this topic (Pag: 11, line: 359-363).
Furthermore, a study conducted based on the same surveillance system data (PASSI) in Tuscany region, which adopted a specific module on health literacy using HLS-EU-Q6 questionnaire, highlighted that health literacy did not affect the likelihood of influenza vaccination uptake among high-risk groups for influenza.(Zanobini, P.; Lorini, C.;Caini, S.; Lastrucci, V.; Masocco, M.;Minardi, V.; Possenti, V.; Mereu, G.;Cecconi, R.; Bonaccorsi, G. Health Literacy, Socioeconomic Status and Vaccination Uptake: A Study on Influenza Vaccination in a Population-Based Sample. Int. J. Environ. Res. Public Health 2022, 19, 6925. https://doi.org/10.3390/ijerph19116925)
Comments on the Quality of English Language
Minor editing required. English is good overall.
Reviewer 2 Report
Comments and Suggestions for Authors
This paper is an interesting work, because it deals with Health disparities with immigrants. Nevertheless before publication, I suggest some minor changes that can improve the manuscript.
1. Please explain the acronyms of PASSI the first time that is used.
2. Also, explain the difference between the two systems. PASSI and PASSI D'Argento
3. Are these databases publicly available? Please explain. If yes, provide a link.
4. In Table 2, there is data that is in bold, which apparently means significant findings. Please incude at the foot of the table a note indicating it.
5. Use the terminology in a consistent way. For example the Title of Table 2, speaks of RRR , while the body of the table includes OR.
6. Consistently use decimals, Table 2 has 3 decimals, while in the results section the numbers have 2 decimals.
7. , I suggest to the authors to use a DAG (Directed acyclic graph) to explain the theoretical relations between variables. You can do that easily usin the daggity software https://dagitty.net/
Author Response
Review 2
( ) I would not like to sign my review report
(x) I would like to sign my review report
Quality of English Language
( ) I am not qualified to assess the quality of English in this paper
( ) English very difficult to understand/incomprehensible
( ) Extensive editing of English language required
( ) Moderate editing of English language required
( ) Minor editing of English language required
(x) English language fine. No issues detected
|
Yes |
Can be improved |
Must be improved |
Not applicable |
|
|
Does the introduction provide sufficient background and include all relevant references? |
( ) |
( ) |
(x) |
( ) |
|
Are all the cited references relevant to the research? |
(x) |
( ) |
( ) |
( ) |
|
Is the research design appropriate? |
( ) |
( ) |
(x) |
( ) |
|
Are the methods adequately described? |
( ) |
( ) |
(x) |
( ) |
|
Are the results clearly presented? |
( ) |
( ) |
(x) |
( ) |
|
Are the conclusions supported by the results? |
( ) |
(x) |
( ) |
( ) |
Comments and Suggestions for Authors
This paper is an interesting work, because it deals with Health disparities with immigrants. Nevertheless before publication, I suggest some minor changes that can improve the manuscript.
- Please explain the acronyms of PASSI the first time that is used.
R: we added the first time that we used the acronyms the meaning (Pag: 2, line: 84-85).
- Also, explain the difference between the two systems. PASSI and PASSI D'Argento
R: The two surveillance systems used in this paper, both coordinated by the Istituto Superiore di Sanità, PASSI and PASSI d’Argento are focused on the adult population (18-69 year-olds) and the elderly (people aged 65 years and over), respectively. We added in 2.1. PASSI and PASSI D'Argento data collection section some details on the two systems highligthing the changes in the text.
- Are these databases publicly available? Please explain. If yes, provide a link.
R: The data presented in this study are available on request from the corresponding author. The data are not publicly available due to restrictions, such as privacy or ethical as reported in the Data Availability Statement.
- In Table 2, there is data that is in bold, which apparently means significant findings. Please incude at the foot of the table a note indicating it.
R: Thank you, we changed it accordingly both in Table 2 and Table 3.
- Use the terminology in a consistent way. For example the Title of Table 2, speaks of RRR, while the body of the table includes OR.
R: Thank you, there was an oversight. We used correctly RRR both in Table 2 and Table 3.
- Consistently use decimals, Table 2 has 3 decimals, while in the results section the numbers have 2 decimals.
R: We uniformed both tables using two decimals like in the results section.
- I suggest to the authors to use a DAG (Directed acyclic graph) to explain the theoretical relations between variables. You can do that easily usin the daggity software https://dagitty.net/
R: Thank you for this helpful suggestion which we will surely use in the future. However, for this study, our focus is to identify differences in the propensity of the foreign resident population in Italy and the autochthonous population to be vaccinated evaluating potential predictors investigated with the short COVID-19 module, that we added during pandemic period to the standard questionnaire, to explore main several COVID-19-related patterns such as perceptions, experience with the infection, accessibility to care, trust in the ability of the healthcare system to manage the emergency. So basically our study is a cross-sectional survey, with the aim to describe the association between willingness to be vaccinated and several factors by the two subsamples, for this reason we think the use of DAG is unsuitable in this context. Moreover, we checked the best multivariate model using the stepwise procedure.
Reviewer 3 Report
Comments and Suggestions for Authors
The study was conducted using survey approach of what was classified as Italian born versus foreign born residents living in Italy. The objectives are to determine potential differences in vaccine hesitancy based on these two categories and then further evaluate determinants of any difference. The study is well designed using acceptable and valid epidemiological methodology. There are a number of concerns I have as listed below.
The term "migrant" although accurate I belive should be recast as "foreign born' versus born in Italy. It's a picky point but migrant term has implications of a transient residents status whereas in reality these foreign born respondents have lived in Italy for 10 or more years and are acculturated as they can at least speak adequate Italian to complete a survey. This leads to another issue in not addressing why foreign born residents would differ in their vaccine hesitancy versus native born. How does Italian societal structure promote vaccine hesitancy in this group? Although the study attempts to uncover determinants of differences such as socio economic and mistrust- these are almost universal in multiple studies particularly in the US. And the reported outcomes show no associated risk for these determinants except being female. In the US institutional racism and historical distrust of the government by African Americans is known to promote vaccine hesitancy and is this a driver in Italian society? It would have been fascinating to include a few questions to attempt to tease out these issues in the foreign born versus native Italians.
The sample size is very small in the foreign born cohort and as such most of the determinant analysis is not significant with large CI. Utilizing a futility index calculation would be helpful to determine how valid the vaccine hesitancy outcomes are based on the small numbers.
The authors point out there was no classification of where the foreign born residents were from. This would seem to be critical as a large migration of agricultural-industrial-service focused residents may have vastly different insight into the clinical implications of vaccination.
I belive the survey responses were also incomplete as it did not include an "unsure" response. It was clear in the pandemic that people were often not certain on a decision based on often conflicting data and information from government officials and as such would be a cohort that could potentially be influenced by public health information that is accurate.
What is lacking in so many of these similar studies are simple questions related to the actual reason for a person to reject or accept vaccine or to remain unsure. This study also suffers from this short coming. Grading the importance of a reason for a decision - such as "which of the following is the primary reason for your decision" then give alternatives in rank order would have been highly innovative and perhaps tease out the difference in the foreign born versus native Italian born cohorts. As it stands I only know that if someone takes a flu vaccine likely they will take a covid vaccine no matter where they where born.
In conclusion the project made a valid attempt to understand if there were difference in vaccine hesitancy in two resident cohorts and further determine what may be the determinants of the differences. Unfortunately we are not certain what could be the reason or reasons for these differences but may have important implications for Italian public health measures in the future.
Comments on the Quality of English LanguageOverall quite good- Some editing in a number of sentences required and issues with punctuation.
Author Response
Review 3
( ) I would not like to sign my review report
(x) I would like to sign my review report
Quality of English Language
( ) I am not qualified to assess the quality of English in this paper
( ) English very difficult to understand/incomprehensible
( ) Extensive editing of English language required
(x) Moderate editing of English language required
( ) Minor editing of English language required
( ) English language fine. No issues detected
|
Yes |
Can be improved |
Must be improved |
Not applicable |
|
|
Does the introduction provide sufficient background and include all relevant references? |
(x) |
( ) |
( ) |
( ) |
|
Are all the cited references relevant to the research? |
(x) |
( ) |
( ) |
( ) |
|
Is the research design appropriate? |
( ) |
(x) |
( ) |
( ) |
|
Are the methods adequately described? |
(x) |
( ) |
( ) |
( ) |
|
Are the results clearly presented? |
(x) |
( ) |
( ) |
( ) |
|
Are the conclusions supported by the results? |
( ) |
( ) |
(x) |
( ) |
Comments and Suggestions for Authors
The study was conducted using survey approach of what was classified as Italian born versus foreign born residents living in Italy. The objectives are to determine potential differences in vaccine hesitancy based on these two categories and then further evaluate determinants of any difference. The study is well designed using acceptable and valid epidemiological methodology. There are a number of concerns I have as listed below.
The term "migrant" although accurate I belive should be recast as "foreign born' versus born in Italy. It's a picky point but migrant term has implications of a transient residents status whereas in reality these foreign born respondents have lived in Italy for 10 or more years and are acculturated as they can at least speak adequate Italian to complete a survey. This leads to another issue in not addressing why foreign born residents would differ in their vaccine hesitancy versus native born. How does Italian societal structure promote vaccine hesitancy in this group? Although the study attempts to uncover determinants of differences such as socio economic and mistrust- these are almost universal in multiple studies particularly in the US. And the reported outcomes show no associated risk for these determinants except being female. In the US institutional racism and historical distrust of the government by African Americans is known to promote vaccine hesitancy and is this a driver in Italian society? It would have been fascinating to include a few questions to attempt to tease out these issues in the foreign born versus native Italians.
R: We thank for the suggestion for term “migrant”, we corrected in the abstract section an oversight in line with our study definition in which foreign residents are the respondents residing in Italy with foreign citizenship. We want to precise that when “migrant” was used, in the introduction or discussion sections, is referred to the findings of cited references.
Although Italian National Health system is universal public healthcare offering care and prevention to the whole population present in the state (also transient residents), it is well known that there are inequalities in access to health services due to socioeconomic and cultural characteristics. However, we cannot exclude access barriers linked to race, but unfortunately in our surveillances we did not have information on these topics.
We agree with your suggestion to include a few questions to attempt these issues but a specific new and ad hoc survey is needed and could be considered in future studies.
The sample size is very small in the foreign born cohort and as such most of the determinant analysis is not significant with large CI. Utilizing a futility index calculation would be helpful to determine how valid the vaccine hesitancy outcomes are based on the small numbers.
R: Thank you for the suggestion, but we have never known the application of this index in observational studies like ours. We found that futility index is crucial in clinical trials to prevent unnecessary exposure of participants to potentially ineffective treatments. We will certainly deep this topic in the future.
The authors point out there was no classification of where the foreign born residents were from. This would seem to be critical as a large migration of agricultural-industrial-service focused residents may have vastly different insight into the clinical implications of vaccination.
R: We agree with this observation and believe that it would have been very interesting to deep the analysis by country of origin of foreigners, since we had this information. Unfortunately, we didn’t perform this analysis due to the small subsample size.
I belive the survey responses were also incomplete as it did not include an "unsure" response. It was clear in the pandemic that people were often not certain on a decision based on often conflicting data and information from government officials and as such would be a cohort that could potentially be influenced by public health information that is accurate.
R: We appreciate your input, but we made an a priori decision to exclude from our analysis the sample interviews where respondents answered "I don't know" to the question, "If a vaccine against COVID-19 were available, would you be willing to be vaccinated?" This decision was based on our belief that incorporating this subgroup could have heightened the uncertainty of our results. Nevertheless, we conducted a sensitivity analysis on this subgroup to assess whether the characteristics of these interviewees differed from the main sample and to evaluate if this exclusion might introduce bias into our results.
What is lacking in so many of these similar studies are simple questions related to the actual reason for a person to reject or accept vaccine or to remain unsure. This study also suffers from this short coming. Grading the importance of a reason for a decision - such as "which of the following is the primary reason for your decision" then give alternatives in rank order would have been highly innovative and perhaps tease out the difference in the foreign born versus native Italian born cohorts. As it stands I only know that if someone takes a flu vaccine likely they will take a covid vaccine no matter where they were born.
R: We agree with this comment and we believe that investigation of the rejection or acceptance vaccine reasons could be very important. Unfortunately, the data used in this study comes from a standardized surveillance systems with consolidated questionnaire since 2008 with more than 80 questions and with average interview duration of 30 minutes. So we think that, for deep the reasons behind the willingness to be vaccinated for COVID, a specific new and ad hoc survey is needed; although we added during pandemic period to the standard questionnaire a short COVID-19 module to explore main several COVID-19-related patterns such as perceptions, experience with the infection, accessibility to care, trust in the ability of the healthcare system to manage the emergency, as reported in the paper.
In conclusion the project made a valid attempt to understand if there were difference in vaccine hesitancy in two resident cohorts and further determine what may be the determinants of the differences. Unfortunately we are not certain what could be the reason or reasons for these differences but may have important implications for Italian public health measures in the future.
R: We thank for the comment that address further fields of investigation.
Comments on the Quality of English Language
Overall quite good- Some editing in a number of sentences required and issues with punctuation.
Round 2
Reviewer 1 Report
Comments and Suggestions for Authors
I thank the Authors for integrating my comments and suggestions.
Comments on the Quality of English LanguageMinor editing of English language required
Reviewer 3 Report
Comments and Suggestions for Authors
Compliments to the authors on an excellent revised manuscript